# What Matters in Autonomous Driving Anomaly Detection: A Weakly Supervised Horizon

**Abstract.** Video anomaly detection (VAD) in autonomous driving scenario is an important task, however it involves several challenges due to the ego-centric views and moving camera. Due to this, it remains largely under-explored. While recent developments in weakly-supervised VAD methods have shown remarkable progress in detecting critical real-world anomalies in static camera scenario, the development and validation of such methods are yet to be explored for moving camera VAD. This is mainly due to existing datasets like DoTA not following training pre-conditions of weakly-supervised learning. In this paper, we aim to promote weakly-supervised method development for autonomous driving VAD. We reorganize the DoTA dataset and aim to validate recent powerful weakly-supervised VAD methods on moving camera scenarios. Further, we provide a detailed analysis of what modifications on state-of-the-art methods can significantly improve the detection performance. Towards this, we propose a "feature transformation block" and through experimentation we show that our propositions can empower existing weakly-supervised VAD methods significantly in improving the VAD in autonomous driving.

**Keywords:** Video Anomaly Detection · Weakly-supervised Learning

## 1 Introduction

Anomaly detection on egocentric vehicle videos is a prominent task in computer vision to ensure safety and take actionable decision (such as emergency breaking) in a autonomous driving scenario. While video anomaly detection (VAD) in static CCTV scenarios has been extensively studied in recent research, egocentric vehicle view anomaly detection (ego-VAD) remains largely unexplored. This is due to the complexity involved in ego-VAD as it poses several unique challenges. These include: (i) complex dynamic scenarios due to moving cameras, (ii) low camera field of view, (iii) little to no prior cues before anomaly occurrence. Furthermore, the previous methods majorly focused on pixel reconstruction based unsupervised anomaly detection while a few follow supervised settings. However, the unsupervised methods [1, 2, 17, 20] have low generalization ability to diverse scenarios and tends to generate false positives for minor variations from training samples. Anomalies are measured against a contextual notion of normalcy which changes from region to region in traffic scenarios, which poses a unique problem for unsupervised techniques. On the contrary, supervised methods have moderate generalization ability in the diverse real-world scenarios but obtaining the

full temporal annotation required for the training of these models is laborious and time consuming.

To combat this, recent static camera VAD approaches [6,11,18,21,25–27,29] adopt a weakly-supervised binary classification paradigm where both normal and anomaly videos are used during training. In this setting, for a long untrimmed video sequence, only coarse video-level labels (*i.e. normal and anomaly*) are required for training instead of frame-level annotations. Here, previous approaches first extract features using a pre-trained frozen off-the-shelf feature backbone (*i.e.* 3D ConvNet, video transformer) and then learn an MLP ranker by multiple instance learning (MIL) based optimization. Largely, previous methods consider only global feature representation (i.e. features extracted from the whole frame) for optimizing the MLP ranker. However some specialized methods extract both global and local features to promote subtle and sharp VAD. Furthermore, for optimizing the MLP ranker, earlier WSVAD approaches adapt a classical MIL loss proposed by [18] which selects two instances based on the presence of abnormality (*i.e. one each from normal and anomaly videos*) to take part in the optimization process. Recent popular weakly-supervised VAD methods [3,19,28] follow a magnitude-based optimization wherein they encourage the sharp abnormal cues of short anomalies to take part in optimization. This feature magnitudes-based optimization is influenced by strong spatio-temporal variation across temporal segments leading to effective separability for sharp and global anomalies.

Another, critical aspect of weakly-supervised VAD lies in effective temporal modeling to discriminate anomalies from normal events. To promote this, previous classical methods [18,21] adopt conventional temporal modeling networks like TCN [10], LSTM [15] to discriminate short anomalies from normal events. In contrast, authors in [19] proposed a multi-scale temporal convolution network (MTN) for global temporal dependency modeling between normal and anomaly segments. Recently, Zhou *et al.* [28] and Chen *et al.* [3] adopt transformer-based global-local and focus-glance blocks respectively to capture long and short-term temporal dependencies in normal and anomalous videos. Distinctively, Majhi *et al.* [14]propose a Outlier-Embedded Cross Temporal Scale Transformer (OE-CTST) that first generates anomaly-aware temporal information for both long and short anomalies and hence allows the transformer to effectively model the global temporal relation among the normal and anomalies. Recent weakly-supervised VAD methods empowered by effective temporal modeling ability and strong optimization ability with limited supervision have gained popularity in static camera condition, however their adaptation to moving ego-camera setting is still unexplored.

Motivated by this, in this paper we aim to provide an extensive exploration of recent popular weakly-supervised methods on ego-centric VAD task. We choose four state-of-the-art (SoTA) reproducible methods: RTFM [19] (ICCV'21), MGFN [3] (AAAI'23), UR-DMU [28] (AAAI'23), and OE-CTST [14] (WACV'24) for quantitative and qualitative analysis. Further, as weakly-supervised methods majorly relay on pre-computed input feature maps, we leverage recent popular vision-language model CLIP [16] for backbone feature extraction. Next, we proceed to propose a feature transformation block (FTB) to enhance the temporal saliency which can enable better temporal modeling and optimization with feature magni-

tude supervision in SoTA methods. Further, as existing ego-centric VAD datasets like DoTA [23] does not have normal samples in training split, so the official DoTA dataset is not useful for weakly-supervised training (requires both normal and anomaly samples for training). Thus, we reorganize the training split of DoTA dataset to fulfill the weakly-supervised training regime and kept the test split as in official DoTA dataset for fair comparison with previous unsupervised methods. We declare this reorganize DoTA dataset as WS-DoTA to promote weakly supervised research exploration on ego-cetric VAD task. Through experimentation, we have shown in section 5 that *what matters in weakly-supervised learning of anomalies in ego-centric autonomous driving videos*. Further, we show how the proposed FTB enhances the SoTA methods performance significantly on WS-DoTA dataset.

## 2    Preliminaries of Video Anomaly Detection in Weakly-Supervised Setting

Video anomaly detection (VAD) aims to detect whether an anomaly is occurring at the current moment ($t$). For VAD, an algorithm can compute an anomaly score $s(t)$ for the current frame $f_t$. In the context of supervised anomaly detection, a classifier needs full temporal annotations of each frame in videos. However, obtaining temporal annotations for long videos is time

**Table 1:** WS-DoTA Dataset Statistics. The numbers in red denote the statistics for only the abnormal segment of the videos. Here, abnormal classes in test splits are **ST:**Collision with another vehicle which starts, stops, or is stationary, **AH:** Collision with another vehicle moving ahead or waiting, **LA:** Collision with another vehicle moving laterally in the same direction, **OC:**Collision with another oncoming vehicle, **TC:**Collision with another vehicle which turns into or crosses a road, **VP:** Collision between vehicle and pedestrian, **VO:** Collision with an obstacle in the roadway, **00:** Out-of-control and leaving the roadway to the left or right

| Frame Count | Train Split | | Test Split | | | | | | | |
|---|---|---|---|---|---|---|---|---|---|---|
| | Normal | Anomaly | ST | AH | LA | OC | TC | VP | VO | OO |
| Average | 737.8 | 104.6 | 25.5 | 32.6 | 36.7 | 28.4 | 29.1 | 30.1 | 30.4 | 49.2 |
| Minimum | 287 | 30 | 9 | 7 | 4 | 5 | 1 | 10 | 12 | 9 |
| Maximum | 750 | 299 | 50 | 84 | 158 | 203 | 135 | 71 | 75 | 143 |
| Total Videos | 3592 | 2689 | 24 | 164 | 168 | 115 | 390 | 35 | 29 | 106 |

consuming and laborious. Weakly-supervised setting relaxes the assumption of having these accurate temporal annotations. Here, only video-level labels indicating the presence of an anomaly in the whole video is needed. A video containing anomalies is labeled as positive and a video without any anomaly is labeled as negative. Formally the weakly-supervised anomaly detection task can be formulated as:

Given a set of normal , anomaly untrimmed video for training and a test query untrimmed video $V$ with $n$ frames *i.e.* $V = \{f_1, f_2, f_3, \ldots, f_n\}$, goal is to find out a set of $m$ ($m \leq n$) frames, $V_{anomaly}$ that contains an anomaly video pattern *i.e.* $V_{anomaly} = \{f_1^a, f_2^a, f_3^a, \ldots, f_m^a\}$, where $V_{anomaly} \subseteq V$.

- $V_{anomaly}$ can be $\phi$, if all frames of $V$ are normal.
- $V_{anomaly}$ can be $V$, if all frames of $V$ contain anomaly.

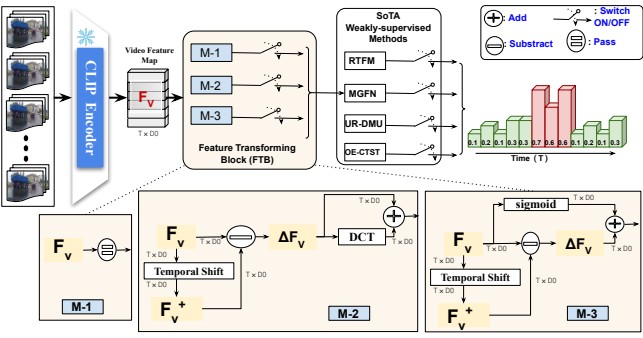

**Fig. 1:** Our Framework for experimental analysing of Weakly-supervised video anomaly detection methods on autonomous driving videos. Here, we integrate a feature transformation block (FTB) to improve state-of-the-art methods perfromance.

## 3    WS-DoTA Dataset

To train wekly-supervised models we require the dataset to contain both normal and anomalous videos. We curate a suitable dataset having over 6000 videos for training, and over 1000 videos for testing all of which are anomalous. The training split contains videos from Detection of Traffic Anomaly (DoTA) **??** which contains anomalous videos and $D^2$-City dataset which contains normal videos. The test split contains videos from only the DoTA dataset.

## 4    Benchmark Methods Discussion and Our Proposition

The performance of weakly-supervised VAD algorithms keeps improving with the recent state-of-the-art (SoTA) methods obtaining impressive results on publicly available benchmark datasets. For this, we have analyzed four SoTA methods and their functionality on ego-centric vehicle view moving camera dataset. A typical framework for analyzing SoTA methods can be seen in Figure 2. The functional analysis begins with extracting `off-the-shelf` video features from CLIP [16] image encoder $F_v \in \mathbb{R}^{T \times D0}$, where $T$ and $D0$ is the temporal (*i.e.* no.of frames) and embedding dimension respectively. Next, the $F_v$ is spatio-temporally enhanced via a proposed "feature transformation block (FTB)" and the resultant is passed it to SoTA methods for learning the abnormality. The functional analysis framework in Figure 2 is designed such that by "`switching on`" a particular SoTA method, the respective anomaly detection performance is reported. Next we briefly characterize the SoTA methodologies before proceeding for the description of proposed FTB. Detailed functional framework of SoTA methods is provided in **supplementary material**.

### 4.1    RTFM [19]

Robust temporal Feature Magnitude (RTFM) addresses one of the major challenge of WSVAD *i.e.* how to localise anomalous snippets from a video labelled as abnormal. The challenge arises due to two reasons: (i) the majority of snippets from an abnormal video consist of normal events, which can overwhelm the training process and challenge the fitting of the few abnormal snippets; (ii) the distinction between normal and abnormal snippets may be subtle, making it difficult to clearly differentiate between the two.

RTFM uses the temporal feature magnitude of video snippets, with low-magnitude features indicating normal (negative) snippets and high-magnitude features indicating abnormal (positive) snippets. It is based on the top-k multiple instance learning (MIL) approach, which involves training a classifier using the k highest-scoring instances from both abnormal and normal videos. Additionally, to capture both long and short-range temporal dependencies within each video, RTFM integrates a pyramid of dilated convolutions with a temporal self-attention module. This combination allows for more comprehensive learning of temporal patterns across different time scales.

### 4.2  MGFN [3]

Magnitude-Contrastive Glance-and-Focus Network (MGFN) advances the notion of RTFM [19] and provides a contrastive learning framework for WSVAD. Using global-to-local information integration mechanism similar to human vision system for detecting anomalies in a long video, MGFN first glances the whole video sequence to capture long-term context information, and then further addresses each specific portion for anomaly detection. Instead of merely fusing spatio-temporal features, the MGFN strategy allows the network to first gain an overview of the scene, then detect scene-adaptive anomalies using global knowledge as a prior. Crucially, unlike the RTFM loss, which simply separates normal and abnormal features without accounting for different scene attributes, they propose a Magnitude Contrastive loss to learn a scene-adaptive cross-video magnitude distribution.

### 4.3  URDMU [28]

To enhance anomaly detection under weak supervision, URDMU uses dual memory units with uncertainty regulation to store and differentiate normal and abnormal prototypes, unlike previous methods that use a single memory for normality. The anomaly memory bank gathers information from anomalous videos, while the normal memory bank learns patterns from normal and abnormal videos.

Building on RTFM [19] finding that normal features typically have low magnitudes, URDU observes normal feature fluctuations due to factors like camera switching. These are modeled with a Gaussian distribution, using a normal data uncertainty learning scheme to create a latent normal space, helping to separate normal and anomalous instances and minimize false alarms. Additionally, a Global and Local Multi-Head Self Attention module is used in the Transformer network to capture video associations more effectively.

### 4.4  OE-CTST [14]

The Outlier Embedded Cross Temporal Scale Transformer (OE-CTST) takes inspiration from transformer-based methods like UR-DMU [28] and MGFN [3]. It proposes a novel framework with an outlier embedder (OE) and a cross temporal scale transformer (CTST). Unlike traditional position embeddings, the OE generates anomaly-aware temporal position encodings by learning from a uni-class distribution, treating outliers as anomalies. These encodings are integrated with temporal tokens and processed by the CTST.

The CTST effectively encodes global temporal relations among normal and abnormal segments through two main components: a multi-stage design and a Cross Temporal Field Attention (CTFA) block. The multi-stage design enables the CTST to examine anomaly-aware tokens at different scales via multi-scale tokenization.

### 4.5   Proposed Feature Transformation Block (FTB)

Primarily, this section considers a new feature transformation strategy ideal for image models like CLIP [16] and weakly-supervised VAD. A key drawback in CLIP for video feature extraction is that it extracts per-frame features as a results it ignores the underlying motion of the video. This underlying motion cue is a relevant attribute in autonomous VAD. Hence a motion enhanced feature map that can highlight the salient temporal region is desirable. For this, we study and propose three modules (M1, M2, M3) of feature transformation as described below.

**M1: Spatial Feature**  As shown in Figure 2, this module considers the raw spatial video feature obtained from Image encoder of CLIP $F_v \in \mathbb{R}^{T \times D0}$ as a baseline. The feature map $F_v$ has enriched spatial semantics thanks to large-scale vision-language pre-training. However, $F_v$ without motion representation alone may not be self-sufficient to represent an abnormality in autonomous VAD.

**M2: Frequency aware Temporal Regularity Feature**  To overcome the lacuna of M1, this module shown in Figure 2 explicitly encodes the motion representation via the temporal regularity feature map $\Delta F_v \in \mathbb{R}^{T \times D0}$ and it's corresponding discrete cosine transform (DCT) coefficients. To obtain $\Delta F_v$, at first, a `temporal shift` operation is applied to $F_v$ that principally moves the temporal feature vector along the temporal dimension. The outcome of the `temporal shift` operator is also a $T \times D0$ dimensional video feature map $F_v^+$ where the first and last temporal tokens are padded and truncated respectively. Then, an absolute difference between $F_v$ and $F_v^+$ is performed to compute the temporal regularity $\Delta F_v$. This operation enables to capture the amount of change between consecutive segments. Further, to enhance the motion representations, discrete cosine transform s applied on top of temporal regularity feature $\Delta F_v$. The motivation and intution behind using DCT for feature enhancement is quite straight forward, as DCT components can represent entire temporal motion sequence and can be sensitive to subtle motion patterns as well. Further, Low-frequency DCT coefficients reflect the movements with steady or static motion patterns, which are not discriminative enough. Thus, we element-wise `added` the resultant of DCT and $\Delta F_v$ to infuse low-frequency component of DCT with higher order temporal regularity feature. This feature transformation allow the sharp temporal regularity feature to be aware of subtle low-frequency features which is to be used by SoTA method for anomaly separability learning.

**M3: Spatial aware Temporal Regularity Feature**  As shown in Figure 2, this module extends the notion of M2 in feature enhancement and re utilizes the enriched vision-language spatial semantics on top of temporal regularity feature $\Delta F_v$. The motion salient temporal regularity features has the ability to capture sharp changes however it's agnostic about spatial scenario variances. Moreover, these spatial information could be critical along with the motion encoding in autonous deiving condition where the scenario is quite dynamic. Thus, In order to complement the temporal regularity features $\Delta F_v$ via spatial feature, $F_v$ is `sigmoid` activated and `added` element-wise to $\Delta F_v$ to result in a spatial aware temporal regularity feature map to be used by SoTA methods.

## 5   State-of-the-art Quantitative Comparison and Qualitative Analysis

In Table 2, we compare the four popular weakly-supervised state-of-the-art (W-SoTA) methods RTFM [19], MGFN [3], URDMU [28], and OE-CTST [14] with the classical unsupervised and supervised methods. Further, we analyse the effectiveness of our feature transformation block (FTB) in performance gain across four W-SoTA method. To suuport this, a detailed qualitative analysis is shown in Figure 2. In Table 2, the performances are compared across two indicator *i.e.* overall and class-wise performance. Kindly note that, unlike unsupervised methods we only use raw RGB frames as input to W-SoTA methods, hence it is fair to compare the results on only RGB modalities. Additional qualitative analysis is provided in **supplementary material**.

**Overall Performance (AUC%)** In this indicator, RTFM [19] with **M3** feature transformation (*i.e.* spatial aware temporal regularity feature) has a significant performance gain of +10.7% compared to unsupervised Anopred [13] method and further, it surpasses the fully supervised TRN by +0.2%. Similar impressive performance gains are also achieved by other W-SoTAs with **M3** feature transformation, which shows the effectiveness of **M3** in highlighting anomaly relevant salient clues in the input feature maps. In contrast,

**Table 2:** State-of-the-art comparisons on the test set of WS-DoTA dataset across overall and class-wise performance indicator, where the considered test-set of WS-DoTA has the same test protocol as DoTA [23] dataset for fair comparison with previous.

| Methods | Overall AUC (%) | ST | AH | LA | OC | TC | VP | VO | OO |
|---|---|---|---|---|---|---|---|---|---|
| *Unsupervised Method with RGB only Feature* | | | | | | | | | |
| ConvAE (gray) [7] | 64.3 | - | - | - | - | - | - | - | - |
| ConvAE (flow) [7] | 66.3 | - | - | - | - | - | - | - | - |
| ConvLSTMAE (gray) [5] | 53.8 | - | - | - | - | - | - | - | - |
| ConvLSTMAE (flow) [5] | 62.5 | - | - | - | - | - | - | - | - |
| AnoPred (RGB) [13] | 67.5 | 70.4 | 68.1 | 67.6 | 67.6 | 69.4 | 65.6 | 64.2 | 57.8 |
| AnoPred (Mask RGB) [13] | 64.8 | 69.6 | 67.9 | 62.4 | 66.1 | 65.6 | 65.3 | 58.8 | 59.9 |
| TAD (Bbox+ flow) [24] | 69.2 | - | - | - | - | - | - | - | - |
| TAD [24] + ML [9] [12](Bbox+ flow) | 69.7 | 71.2 | 71.8 | 68.9 | 71.3 | 70.6 | 67.4 | 63.8 | 69.2 |
| Ensemble (RGB + Bbox+ flow) | 73.0 | 75.4 | 75.5 | 71.0 | 75.0 | 74.5 | 70.6 | 65.2 | 69.6 |
| *Supervised method with RGB only Feature* | | | | | | | | | |
| LSTM [8] (RGB) | 63.7 | - | - | - | - | - | - | - | - |
| Encoder-Decoder [4] (RGB) | 73.0 | - | - | - | - | - | - | - | - |
| TRN [22] (RGB) | 78.0 | - | - | - | - | - | - | - | - |
| *Weakly-Supervised Methods with M1: Spatial only Feature* | | | | | | | | | |
| RTFM [19] | 57.9 | 59.8 | 58.6 | 57.6 | 56.5 | 56.2 | 55.2 | 51.6 | 60.6 |
| MGFN [3] | 66.6 | 57.1 | 66.2 | 64.6 | 69.6 | 67.0 | 63.0 | 64.3 | 69.3 |
| URDMU [28] | 57.5 | 50.8 | 58.8 | 60.0 | 57.4 | 56.7 | 55.3 | 53.2 | 56.2 |
| OE-CTST [14] | 70.9 | 64.2 | 71.4 | 71.5 | 68.2 | 71.2 | 66.2 | 69.6 | 75.2 |
| *Weakly-Supervised Methods with M2: Frequency aware Temporal Regularity Feature* | | | | | | | | | |
| RTFM [19] | 56.0 | 57.1 | 56.1 | 55.7 | 53.4 | 56.2 | 57.9 | 53.9 | 58.1 |
| MGFN [3] | 67.4 | 67.1 | 70.0 | 66.8 | 67.9 | 67.6 | 67.6 | 73.7 | 69.0 |
| URDMU [28] | 54.8 | 58.4 | 56.3 | 54.3 | 53.0 | 54.7 | 52.8 | 54.5 | 55.1 |
| OE-CTST [14] | 71.9 | 66.3 | 70.6 | 72.0 | 72.1 | 71.1 | 67.1 | 76.4 | 75.9 |
| *Weakly-Supervised Methods with M3: Spatial aware Temporal Regularity Feature* | | | | | | | | | |
| RTFM [19] | 78.2 | 62.7 | 79.2 | 78.7 | 76.5 | 77.5 | 74.7 | 79.8 | 83.1 |
| MGFN [3] | 67.4 | 60.8 | 68.9 | 66.5 | 66.8 | 67.3 | 61.2 | 66.1 | 68.0 |
| URDMU [28] | 73.0 | 63.8 | 71.1 | 72.4 | 72.9 | 74.9 | 65.4 | 79.5 | 75.9 |
| OE-CTST [14] | 75.6 | 63.6 | 77.4 | 76.0 | 73.8 | 74.9 | 73.3 | 76.2 | 78.1 |

W-SoTAs with **M1** feature transformation (*i.e.* spatial only features) performs poorly w.r.t unsupervised and supervised methods. This is mainly due to the lack of motion representative features in CLIP [16] backbone which may turn out crucial in ego-centric video anomaly detection. Further, to encourage motion features in CLIP embeddings, W-SoTAs are analysed with **M2** feature transformations (*i.e.* frequency aware Temporal regularity features). However, in contrast to our assumptions the performance gain by W-SoTAs are marginal or even lower for some cases. From details investigation, we found that the DCT frequency components in **M2** feature transformations has sensitivity even for subtle motions. Thus, it tends to produce many false positives in autonomous driving condition as the dynamic scene has many subtle to sharp motion cues. The drawback of **M1** and **M2** feature transformations are addressed by **M3** by encouraging only sharp mo-

tion cues while retaining the rich spatial semantic of the scene. Thanks to this, all the W-SoTAs considered for analysis has larger performance gain.

**Class-wise Performance (AUC%)** To bring additional analytical insights to W-SoTA performance comparison, Table 2 provides an anomaly class-wise performance comparison. The W-SoTAs with **M3** feature transformation has the significant performance gain many classes with few exceptions like "ST", where W-SoTAs across all feature transformations (M1, M2, M3) are less better than unsupervised Anopred [13] method. However, from empirical investigation we found that "ST" categories have less abnormal samples compared to others in the test set of WS-DoTA and thus W-SoTAs

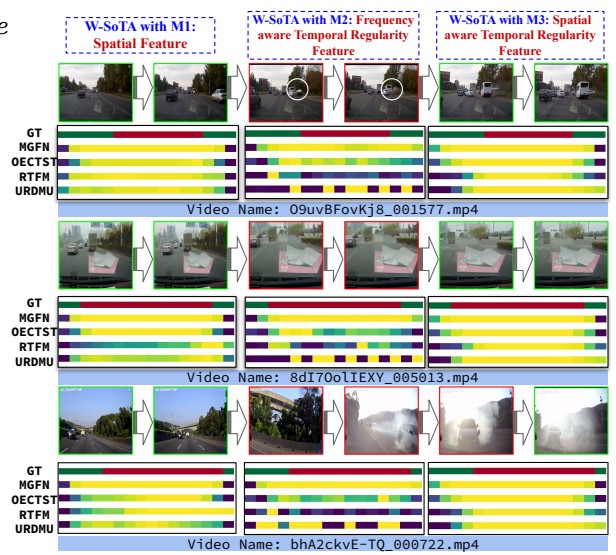

**Fig. 2:** Visualization of Ground truth vs. prediction heatmaps for SoTAs in withe different feature mpas obtained from feature Transformation block (FTB). We portray such visualization for three challenging videos. More visualization can be found in appendix.

with our feature transformation block could not outperform with less test samples. Apart from this, W-SoTAs (specifically RTFM) able to achieve significant performance gain (*i.e.* at least +8% and at most +25% ) in class-wise performance thanks to the **M3** feature transformations where salient sharp motion cues are encouraged along with the relevant spatial semantics.

From state-of-the-art comparison and analysis, it is evident that weakly-supervised methods has the potential to improve the anomaly detection in autonomous driving condition provided the input feature maps has the explicit encoding for motion and spatial semantics cues.

## 6   Conclusion

In this work, we provide a experimental exploration of state-of-the-art weakly-supervised methods on video anomaly detection for autonomous driving scenarios. By covering experimental depth and breadth, it is evident that the weakly-supervised methods along with our feature transformation block has the potential to drive the detection performances far ahead of classical unsupervised methods. Next, to promote subsequent research of weakly-supervised method on autonomous driving video anomaly detection task, we provide a WS-DoTA dataset and the validation of benchmark methods to be considered for baseline. in future, we will develop specialized framework for detection and description of video anomalies in autonomous driving scenario.

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
