# OpenReview forum: "What Matters in Autonomous Driving Anomaly Detection: A Weakly Supervised Horizon"
_thecvf.com/ECCV/2024/Workshop/ROAM — ROAM ECCV 2024 Oral_

### Official Review · Reviewer_aj7V · 2024-08-06
**Review of video anomaly detection for autonomous driving**

**Rating:** 6
**Confidence:** 4

**Review:**

This paper provides an experimental exploration of SOTA weakly-supervised methods on video anomaly detection for autonomous driving.

pros:

1. This paper reorganize the DoTA dataset and propose WS-DoTA, aiming to validate recent methods on moving camera scenarios. It chooses four SOTA methods: RTFM, MGFN, UR-DMU and OE-CTST for quantitative and qualitative analysis.
2. It leverages the feature from CLIP and extends to temporal modeling using a Feature Transformation Block (FTB). Experiments have demonstrated the effectiveness of the proposed method.
3. The visualization in Figure 2 very clearly and intuitively showcases the characteristics of W1, W2, and W3.

cons:

1. The exploration of the experiments is somewhat insufficient. Since it is a video task, it would be better to use some video models' pre-trained features. Of course, CLIP features have better semantic qualities, which might provide a significant advantage in anomaly detection. However, the experiments lack discussion on this point.
2. The motivation of the new benchmark is to explore performance under moving camera conditions. However, the paper lacks further discussion on this topic, such as which anomaly detection methods suitable for static cameras show significant degradation under moving camera conditions.
3. some writing issues: cite error in L136, lack of explanation of colors used in Table2, wekly in L133.

---

### Official Review · Reviewer_RDxp · 2024-08-12
**Review of Autonomous Driving Anomaly Detection**

**Rating:** 7
**Confidence:** 3

**Review:**

This paper proposes a framework for experimental analysis of weakly-supervised video anomaly detection methods on autonomous driving videos, while also reorganizing the DoTA dataset as the experimental dataset.

pros:

1. The paper reorganizes the DoTA dataset to create the WS-DoTA dataset, providing a benchmark for research in weakly-supervised video anomaly detection.

2. The paper also introduces a Feature Transformation Block, and its effectiveness is validated across multiple methods.

3. Extensive experiments have convincingly demonstrated the effectiveness of the research.

cons:

1. There is a geometric correspondence between multiple frames of autonomous driving videos, and utilizing these camera intrinsic and extrinsic parameters would aid in motion modeling; however, this paper does not make use of them.

2. Some writing issues: Line 136: ??; Line 323: withe; Line 338：a experimental; Line 346：in future.

---

### Official Review · Reviewer_TWWS · 2024-08-17
**Review of  Weakly Supervised Autonomous Driving Anomaly Detection**

**Rating:** 4
**Confidence:** 3

**Review:**

The paper proposes a weakly-supervised method for autonomous driving VAD and reorganizes the DoTA dataset for evaluating weakly-supervised VAD methods.

pros:
1.The paper provides a benchmark for the WS-DoTA task.
2.The paper proposes a “feature transformation block” and shows the block can empower existing weakly-supervised VAD methods.

cons:
1.The proposed method is relatively simple and has limited innovation, and the experiments is insufficient.
2.The Table2 is a bit confusing. The overall performance is same for MGFN+M2 and MGFN+M3, while the class-wise performance of MGFN+M2 is obviously better?
3.In this paper, the Temporal Regularity Feature is directly collected using the front view images, while, for autonomous driving, depth estimation and 3D pre-training for sequential images may be helpful for motion estimation.

---

### Official Review · Reviewer_9uS5 · 2024-08-21
**This paper introduces an enhanced version of the DoTA dataset and a Feature Transformation Block to strengthen temporal modeling, thereby improving weakly supervised VAD methods for autonomous driving.**

**Rating:** 7
**Confidence:** 3

**Review:**

Pros:
1.The paper addresses the improvement of the SOTA VAD methods by considering enhancements from both data and model structure perspectives. 2. By reorganizing the dataset to better suit weakly-supervised methods, the authors demonstrate a resourc-e and labor-efficient approach. 3. The addition of the temporal Feature Transformation Block, effectively improves the performance of existing methods by incorporating temporal dynamics.

Cons:
1.The paper lacks a thorough explanation and rationale for how the newly constructed dataset addresses the challenges specific to the moving camera scenario. 2. The paper does not consider leveraging pretrained video features. 3. There is insufficient discussion on the limitations. 4. There are some writing errors present, such as on line 136.

---

### Official Review · Reviewer_bkTE · 2024-08-21

**Rating:** 7
**Confidence:** 3

**Review:**

This paper introduces a "feature transformation block" that enhances VAD methods in autonomous driving. Moreover, they benchmark state-of-the-art VAD algorithms and highlight their three proposed modules for feature transformation.
Pros:
1. Authors conducted a compressive benchmark to compare their method against
2. Feature transformations draws ideas from established work like CLIP
3. Demonstrates convincing increase in performance among models

Cons:
1. Originality of work could be expanded from other vision models

Overall a well-written paper.

---

### Decision · Program_Chairs · 2024-08-22

**Decision:**

Accept (Oral)

**Comment:**

The average score of the paper given all the reviews received before the deadline was higher than 5.5 (1 is lowest, 10 is highest), therefore the paper is accepted. The Authors are encouraged to consider feedback for the camera ready version of the paper due on August 31st.